# The Profitability of Technical Analysis during the COVID-19 Market Meltdown

Camillo Lento [1],* and Nikola Gradojevic [2]

1    Faculty of Business Administration, Lakehead University, 955 Oliver Road,
     Thunder Bay, ON P7B 5E1, Canada
2    Department of Economics and Finance, Lang School of Business and Economics, University of Guelph,
     50 Stone Road East, Guelph, ON N1G 2W1, Canada; ngradoje@uoguelph.ca
*    Correspondence: clento@lakeheadu.ca; Tel.: +1-807-343-8387

**Abstract:** This article explores the profitability of technical trading rules around the COVID-19 pandemic market meltdown for the S&P 500 index, Bitcoin, Comex gold spot, crude oil WTI, and the VIX. Trading rule profits are estimated from January to May 2020, including three sub-periods, on a high-frequency data set. The results reveal that the trading rules can beat the buy-and-hold trading strategy. However, only the Bollinger Bands and trading range break-out rules become profitable after transaction costs during the market crash. Moreover, it is found that composite trading signals effectively improve the profitability of technical analysis around the COVID-19 market crash.

**Keywords:** technical analysis; technical trading; market efficiency; COVID-19

## 1. Introduction

With origins dating back to the 1800s, technical analysis is one of the earliest forms of investment analysis. Technical analysts review a time series of past prices (and volumes) to discern patterns that may be useful for predicting future price changes. In general, technical analysis examines the economic value of charting, which attempts to exploit recurring geometric patterns found in the history of prices, and the trading rules (TRs) approach that mechanically applies mathematical rules (or trading indicators) constructed from past and present prices or volumes. Studies that found charting to be profitable include those of Chang and Osler (1999), Lo et al. (2000), and Savin et al. (2007), whereas evidence for the profitability of technical indicators can be found in the work of Levich and Thomas (1993), Brock et al. (1992), Neely et al. (1997), Allen and Karjalainen (1999), Gençay (1998), Ratner and Leal (1999), and Lento and Gradojevic (2007).

Some scholarly contributions combine technical analysis with machine learning and other non-parametric modeling methodologies. For example, Lo et al. (2000) showed that charting based on automatic recognition of the head-and-shoulders or double-bottoms patterns assisted with kernel regressions might provide practical investment value for U.S. stocks. In a related study, Savin et al. (2007) produced risk-adjusted excess returns for the head-and-shoulders price patterns of the S&P 500 and the Russell 2000 stock market indices. Gençay (1998) used simple moving average technical indicators to feed a non-parametric, artificial neural network model, which delivered statistically significant profits when tested against buy-and-hold strategies. In contrast, Allen and Karjalainen (1999) could not find any consistent profitability in the S&P 500 index market when applying genetic programming to search for ex-ante "optimal" TRs. Their method was able to identify only specific low-volatility periods with excess returns over the buy-and-hold strategies. Gradojevic and Gençay (2013) extended a panel of TRs with a fuzzy logic controller and generated technical trading indicators that were more tolerant of uncertainties of trading in financial markets. By doing this, they improved the profitability of simple moving average TRs. They also showed that the profitability of technical indicators was related to market volatility in the

sense that higher volatility was associated with greater (lower) profits from fuzzy (pure) technical trading strategies. More recently, Corbet et al. (2019) tested the profitability of the moving average and trading range break-out technical indicators in the Bitcoin market, but with mixed results. By the same token, Gradojevic et al. (2021) recently employed various machine learning models, such as deep learning neural networks and random forests with technical indicators, as inputs to predict hourly and daily Bitcoin price movements. The results demonstrated that Bitcoin prices were weakly efficient for the hourly but not for the daily horizon. In summary, several papers presented evidence that technical analysis can be profitable and informative for price forecasting. However, its profitability can vary over time, as suggested by the adaptive market hypothesis (Lo 2004).[1]

As is the case in the current study, moving average-based TRs are often used in technical analysis as some of the most successful approaches in practical applications. For instance, Neely et al. (2014) showed that moving average signals could predict equity premium more accurately than some macroeconomic fundamentals. Further, Avramov et al. (2021) documented that the distance between short- and long-run moving averages for a large cross-section of NYSE, AMEX, and NASDAQ stock prices could predict future equity returns. In a related contribution, Zakamulin and Giner (2020) demonstrated that moving average TRs exhibited more robust forecast accuracy in the S&P 500 index market than momentum-based TRs. Similar empirical evidence was also provided by Marshall et al. (2017). Next, Jiang et al. (2017) found robust and substantial evidence of the profitability of TRs for the Chinese stock market. M'ng (2018) devised a novel adjustable moving average indicator for the Far East equity markets that resulted in excess returns over the buy-and-hold strategy. Finally, Kouaissah and Hocine (2020) successfully applied TRs for portfolio selection strategies.

This paper contributes to the above literature by focusing on the performance of TRs during the months surrounding the emergence of the COVID-19 pandemic. Such an investigation is valuable because the research on the performance of TRs during financial crises is relatively scarce. For example, Ni et al. (2013) applied moving average TRs to the BRIC countries' equity markets and found difficulties generating profitability during financial crisis periods. Similarly, Narayan et al. (2013) examined the technical trading profitability of the spot market for commodities and documented that structural breaks generally diminished profits. In contrast, Ivanova et al. (2021) failed to observe significant declines in technical excess returns during financial crises.

To the best of the authors' knowledge, this is the first study to explore the practical usefulness of trend and momentum technical indicators around the time of the COVID-19 pandemic. It not only contributes to the literature on the profitability of TRs during financial crises (e.g., Ivanova et al. 2021), but also provides valuable insights into the market microstructure and traders' behavior in response to such episodes of excess volatility across a variety of asset classes. Technical trading literature typically focuses on a specific market (e.g., as in Neely et al. 1997 or Corbet et al. 2019) or jointly analyzes various market regimes (e.g., Zakamulin and Giner 2020). In contrast, the current paper covers three starkly differential market regimes (pre-pandemic, pandemic crash, and market recovery periods) and essentially tracks the shifts in aggregate traders' beliefs as affected by market volatility. Such an approach offers rich grounds for policy analysis that will be considered in the last section of the paper.

In the current context, the profitability of technical analysis during the novel coronavirus disease (COVID-19) market crash has been a largely unresearched area. The emergence of COVID-19 in Wuhan, China, quickly evolved into a global pandemic declared by the World Health Organization on 11 March 2020. Government responses to the pandemic have been unprecedented, with significant impacts on the global economy and broader society. The COVID-19 pandemic and the resulting government responses sent shock waves throughout financial markets. The main purpose of this paper is to explore the profitability of TRs during the COVID-19 market meltdown and, thereby, to understand the behavior of market participants and the underlying market microstructure during the

market turbulence. More specifically, the paper studies whether excess volatility triggered by the COVID-19 pandemic shocks contributed to the profitability of technical analysis. In addition, the combined signal approach (CSA) is employed and its potential usefulness for increasing profitability at times of excess volatility is analyzed.

The scholarly evidence that concerns the potential links between volatility and technical trading profitability is mixed, and the current paper sheds additional light on this undeveloped strand of the literature. For instance, in a seminal paper, Kho (1996) showed that futures conditional volatility explains an additional 10% of the technical trading profits. Moreover, Han et al. (2013) applied the moving average technical indicator and demonstrated that they could generate large abnormal returns for high-volatility portfolios relative to the CAPM and the Fama and French three-factor models. More recently, Ding et al. (2021) reported cross-sectional profitability for trading strategies based on the VIX. However, a few additional studies on the role of volatility were not as encouraging. For instance, Menkhoff et al. (2012) found that high-interest-rate currencies deliver low returns in times of unexpectedly high volatility, while Menkhoff and Taylor (2007) showed that technical analysis can be more profitable for volatile currencies. Similarly, Hsu et al. (2016) concluded that, after adjusting for risk, there existed no significant relationship between volatility and TRs excess profitability. Finally, Beaupain et al. (2010) indicated that technical trading is riskier in periods of high volatility than in quiet markets.

The rest of the paper is laid out as follows. Section 2 presents the construction of TRs. Section 3 describes the data set, while Section 4 reports the results from individual TRs. Section 5 offers a discussion of the results. Section 6 focuses on the evidence from composite trading signals. Section 7 concludes the paper.

## 2. Methodology

### 2.1. Trading Rules

Three variants of four standard TRs are tested. First, the trend behavior is studied by testing the traditional moving average cross-over (MACO) rule. The paper follows Ratner and Leal (1999) and Gradojevic and Gençay (2013) to differentiate between buy and sell signals based on Equations (1) and (2), respectively.

$$\frac{\sum_{S=1}^{S} r_{i,t}}{S} > \frac{\sum_{L=1}^{L} r_{i,t}}{L} \tag{1}$$

$$\frac{\sum_{S=1}^{S} r_{i,t}}{S} < \frac{\sum_{L=1}^{L} r_{i,t}}{L} \tag{2}$$

where $r_{i,t} = (\ln(P_t) - \ln(P_{t-1}))$ and is the natural log return at the five-minute interval, S is the number of five-minute intervals for the short moving average, and L is the number of five-minute intervals for the long moving average. The paper follows Brock et al. (1992) and Lento (2009) to test the following S and L MACO intervals: (1,50), (1,200), and (5,150).

Second, filter rules (FR) are tested, defined based on a filter size ($f$). Such trading rules are driven by traders exploiting market trends and momentum. Specifically, buy (sell) signals are generated when the log return rises (falls) by $f$ percent above (below) the most recent trough (peak). Prior studies are followed (e.g., Gradojevic and Gençay 2013) and three variants of the FR are tested by defining $f$ as 1%, 2%, and 5%.

Third, the trading range break-out (TRBO) rule is tested. The TRBO generates a buy (sell) signal when the price breaks out above (below) the resistance (support) level. The resistance (support) level is the local maximum (minimum) over the past n days (Brock et al. 1992). Equations (3) and (4) present the TRBO buy and sell signals, respectively.

$$\text{Buy} = P_t > \text{Max}\ \{P_{t-1}, P_{t-2}, \dots, P_{t-n}\} \tag{3}$$

$$\text{Sell} = P_t < \text{Min}\ \{P_{t-1}, P_{t-2}, \dots, P_{t-n}\} \tag{4}$$

where $P_t$ is the price at time t. Again, the prior literature is followed (e.g., Brock et al. 1992) and three variants of the TRBO are tested by calculating the local maximum and minimum based on 50, 150, and 200 5-min intervals. TRBO rules indicate when the balance between demand and supply is settled violently, thus suggesting the initiation or continuation of a directional trend (Kirkpatrick and Dahlquist 2016).

Lastly, the Bollinger Bands (BB) are tested (Bollinger 2001). BBs are traditionally calculated based on an envelope plotted at a two-standard deviation $(+/- 2\sigma)$ distance from the 20-day moving average, thus, denoted by BB (20,2). Consistent with Lento et al. (2007), the profits from BB (20,2) are calculated along with two variants: 30-day moving average $(+/- 2\ \sigma)$ and 20-day moving average $(+/- 1\ \sigma)$. BBs are more of a contrarian trading strategy than the other three TR. Essentially, BBs represent a variation of the trend-following system, i.e., a break-out system that measures range volatility. The trading signal is generated when the price moves out of a channel or band. Hence, it gauges both trend and momentum.

### 2.2. Profitability Measures

Profitability is measured as the returns generated by the trading signals in excess of the buy-and-hold trading strategy (BHTS) returns. The returns from the BHTS are calculated by investing on the first day of the data set, given the trading rule parameters, and holding until the last day. The returns from the TRs are calculated by going long (out of) the market in the five-minute interval after a buy (sell) signal. No notional interest is earned while out of the market. Similar to Gençay (1998), the TR returns are adjusted for an estimated bid-ask spread and brokerage costs. Profits are presented before and after transaction costs as transaction costs can vary significantly among market participants. Transaction costs capture both the bid-ask spread and brokerage trading cost. The bid-ask spread for an exchange-traded fund of an index is used as a proxy for the actual index. Consistent with prior studies (e.g., Ratner and Leal 1999; Lento and Gradojevic 2007), transaction costs are measured as 0.15% of the buy or sell amount.

The Sharpe ratio (SR) is also calculated to provide insights into the risk profile of the profits generated by the trading signals (e.g., Zhu et al. 2015; Gradojevic 2007). Specifically, the SR measures the average excess return (profit) per unit of total risk. The SR is calculated with Equation (5).

$$SR = \frac{\mu - r_f}{\sigma} \tag{5}$$

where $\mu$ is the mean daily return from the trading rule, $r_f$ is the mean daily risk-free based on the Fama-French three-factor model for 2020, and $\sigma$ is the standard deviation of the return series. All three measures are estimated on the entire sample. Note that, all else equal, a larger Sharpe ratio indicates a better trading rule on a risk-return basis.

### 2.3. Bootstrapping Simulations

The statistical significance of the TR profits is estimated with the bootstrap simulation approach developed by Levich and Thomas (1993), whereby an observed series of prices (i.e., the actual data set), with a sample size denoted N + 1, will correspond to a set of N returns. A series of m[th] (m = 1, . . . , M) permutations is randomly generated by reshuffling the N returns from the observed data (M = N!) while holding the start and end points fixed based on their observed values. Each m[th] permutation is random while inheriting the distributional properties of the observed data. Each trading rule will then relate to a unique profit measure (X [m, r]) for each m[th] permutation, thereby generating an empirical distribution of profits. A simulated p-value is estimated by comparing the profits generated by a trading rule on the observed data versus the simulated data.

### 3. Data

A high-frequency data set is employed at the five-minute interval for five asset classes from 1 January 2020 to 12 May 2020. There are a total of 7058 observations for the Bitcoin

(BTC), COMEX gold spot (GLD), NYMEX WTI crude oil (OIL), S&P 500 index (SPX), and the CBOE VIX index (VIX). The data are from the Thomson Reuters Eikon terminal. The returns from these five asset classes could be replicated either through the futures market or exchange-traded funds.[2] Table 1 presents the descriptive statistics of the five-minute interval returns across each dataset.

**Table 1.** Descriptive Statistics.

|  | Mean | Median | Standard Deviation | Skewness | Coefficient of Variation | Excesskurtosis |
|---|---|---|---|---|---|---|
| BTC | 0.005% | 0.007% | 0.624% | −13.29 | 118.57 | 520.33 |
| GLD | 0.002% | 0.002% | 0.163% | −0.05 | 98.62 | 133.50 |
| SPX | −0.001% | 0.002% | 0.342% | −3.34 | 388.83 | 139.00 |
| VIX | 0.021% | 0.000% | 1.510% | 6.29 | 71.69 | 150.25 |
| WTI | −0.008% | 0.000% | 7.415% | −36.01 | 972.99 | 2719.80 |

Notes: Table 1 presents the mean, median, standard deviation, skewness, coefficient of variation, and excess kurtosis for the five-minute interval returns across each of the five markets: Bitcoin (BTC), COMEX gold spot (GLD), NYMEX WTI crude oil (OIL), S&P 500 index (SPX), and the CBOE VIX index (VIX).

Table 1 reveals that the mean and median five-minute returns are close to zero. However, the variation and distribution shape measures suggest many extreme observations for WTI, VIX, BTC, and the SPX markets. This is understandable as the COVID-19 market meltdown was one of the most significant market crashes in decades (based on, e.g., VIX volatility) that also witnessed WTI oil futures prices become negative for the first time in recorded history. The GLD market was the least volatile across the period analyzed, consistent with the idea that gold is a safe-haven investment during times of crisis (e.g., Bredin et al. 2015). The safe-haven characteristic of gold during the pandemic is even more notable as almost all assets expressed in US dollars, including gold, became strongly correlated during this period (Kwapień et al. 2021).

The profitability of the TRs is estimated across the entire period of 1 January 2020, to 12 May 2020. In addition, the profitability of each TR is analyzed across the same 2020 market crash regime shifts identified by Lento and Gradojevic (2021): (i) the normal market regime spans from 1 January to 19 February, just before the SPX crashes, (ii) the market crash regime spans from 20 February to 23 March, representing the peak to trough of the SPX market crash, and iii) the market recovery regime spans from 24 March to 12 May, representing the SPX's recovery from the trough.

The number of trades generated by each TR variant is reviewed across the entire sample and each of the three market regimes (See Appendix A). All TRs generated many signals across the sample, aside from the FR (2%) and FR (5%) in the GLD and SPX markets, which generated fewer signals. The largest number of signals were generated in the VIX market. It is also interesting to note that most TR variants generated the least number of signals during the market crash regime compared to the normal and recovery regimes.

## 4. Results

The daily profits from the TRs across the entire period are presented in Table 2. The results reveal specific concentrations of positive profits both before and after transaction costs on the observed data. For example, at least nine of the 12 variants generated profits before transaction costs in the OIL and SPX markets. Positive profits after transaction costs on the observed data occurred much less frequently across the five asset classes. Moreover, the bootstrapping simulation data suggest that many positive profits are not statistically significant. Interestingly, BBs consistently generated statistically significant profits on the VIX before and after transaction costs.

**Table 2.** TTR Profitability across the full sample.

|  | MACO | MACO | MACO | BB | BB | BB |
|---|---|---|---|---|---|---|
|  | (1,50) | (1,200) | (5,150) | (20,2) | (20,1) | (30,2) |
| **BTC** | −0.32% | −0.22% | −0.47% | **0.21%** | **0.32% *** | **0.13%** |
|  | −1.07% | −0.43% | −0.63% | −0.19% | −0.30% | −0.19% |
| **GLD** | −0.07% | **0.04% *** | **0.04% *** | **0.01%** | **0.01% *** | −0.01% |
|  | −0.78% | −0.22% | −0.13% | −0.41% | −0.62% | −0.38% |
| **OIL** | **2.16%** | **1.22%** | **2.90%** | −1.06% | −0.27% | −0.95% |
|  | **1.49%** | **0.94%** | **2.76%** | −1.43% | −0.85% | −1.22% |
| **SPX** | −0.22% | −0.02% | **0.03%** | **0.27% *** | **0.35% **** | **0.16%** |
|  | −0.93% | −0.37% | −0.15% | −0.13% | −0.27% | −0.14% |
| **VIX** | −2.71% | −0.88% | −0.63% | **1.50% ***** | **1.35% ***** | **0.79% *** |
|  | −3.57% | −1.30% | −0.85% | **1.04% ***** | **0.68% ***** | **0.44% *** |
|  | **FR** | **FR** | **FR** | **TRBO** | **TRBO** | **TRBO** |
|  | **(1%)** | **(2%)** | **(5%)** | **(50)** | **(150)** | **(200)** |
| **BTC** | −0.30% | −0.18% | **0.02%** | −0.22% | **0.24%** | **0.11%** |
|  | −0.54% | −0.26% | **0.01%** | −0.35% | **0.20%** | **0.08%** |
| **GLD** | −0.05% | **−0.01%** | −0.12% | **0.06%** | 0.00% | **0.04% **** |
|  | −0.09% | −0.02% | −0.12% | −0.07% | −0.05% | 0.00% |
| **OIL** | **2.39%** | **3.57%** | **3.36% *** | **2.07%** | **1.15%** | **0.84%** |
|  | **1.60%** | **3.26%** | **3.26% *** | **1.95%** | **1.12%** | **0.81%** |
| **SPX** | **0.10%** | **0.20% **** | **0.28%** | **0.09%** | **0.10%** | **0.17%** |
|  | −0.09% | **0.15%** | **0.28% **** | −0.06% | **0.05%** | **0.14%** |
| **VIX** | −0.51% | −1.09% | −0.94% | −1.05% | −0.32% | **0.21%** |
|  | −2.06% | −1.77% | −1.08% | −1.21% | −0.37% | **0.19%** |

Notes: Table 2 presents the daily profits before and after transaction costs that are located in each cell above and below, respectively. Bold font represents trading profits whereby the trading rule returns exceeded the returns from the naïve buy-and-hold trading strategy (BHTS). Variants of the following trading rules are shown: moving average cross-over (MACO), Bollinger Bands (BB), filter rules (FR), and trading range break-out (TRBO). ***, **, * represent statistically significant trading profits based on the Levich and Thomas (1993) bootstrapping technique at the 1%, 5%, and 10% levels, respectively.

The SRs support the main findings that the largest SRs are clustered within the TRBO rules, which were also the most profitable (See Appendix B). Large TRBO SRs are consistent with similar studies such as Zhu et al. (2015), who found the TBROs were superior on a risk-return basis to moving average rules. Regarding asset classes, the largest SRs are observed in the OIL market relative to the other four markets. Specifically, the SRs were large and positive for all the MACO, FR, and TRBO variants. Only BBs generated low or negative SRs in the OIL market.[3]

Next, the profitability of the TRs is assessed across the three different regimes during the COVID-19 market meltdown (See Appendix C), along with the SRs (un-tabulated). First, the profits across the normal market regime reveal that the TRs consistently generated positive profits only in the OIL market. However, the TRs have generated less positive profits after transaction costs in the OIL market, while the bootstrapping simulation data further reveals that the returns in the OIL market are not statistically significant. The SRs were relatively large in the OIL market. Statistically significant profits were only observed by the two BB variants in the VIX market. In addition, all of the SRs for BBs on the VIX were relatively large, suggesting superiority to the other trading studies on a risk-return basis as well. Overall, the findings during the normal market regime are consistent with the findings across the entire sample.

Next, the majority of trading rule variants generated positive profits on the observed data before transaction costs during the market crash regime. Regarding profits after transaction costs, only the BB and TBRO TRs generated positive profits on the observed data across all five asset classes during the market crash. The bootstrapping simulation data reveal that statistically significant profits were consistently generated during the market crash regime by the BB, FR, and TRBO in the SPX market (SRs were consistently large

except for FR (5%)), the BB and TRBO in the BTC market (SRs were consistently large except for TRBO (50)), and the MACO and TRBO in the GLD market (SRs were consistently large except for MACO (1,50)). These results reveal that the TRBO consistently generated positive profits during the market crash regime.

Lastly, the TRs generated positive profits in the OIL and VIX markets before and after transaction costs across the market recovery regime. The SRs were among the highest for the VIX across all four trading rules, and large in the OIL market for the MACO, FR, and TRBO. The bootstrapping simulation data shows that MACO and FR consistently generated statistically significant profits in the OIL market, which were also accompanied by relatively large SRs.

## 5. Discussion and Analysis

Overall, the analysis reveals that many TRs could generate positive profits on the observed data before transaction costs. However, most of these profits were not robust enough to persist through transaction costs and statistical significance testing (see Table 2). It should be noted that the BBs could generate positive profits before transaction costs on the observed data in the BTC, GLD, SPX, and VIX markets. However, profits were only statistically significant after transaction costs in the VIX market.

The study extends the prior literature that explored the relationship between technical trading models and the implied SPX volatility (i.e., the VIX market). For example, Kozyra and Lento (2011) first highlighted the usefulness of VIX data for calculating technical trading rules. By determining technical trading signals on VIX data while trading on the SPX, they found statistically significant profits, which were more pronounced during periods of high volatility. Various studies have since incorporated the VIX into trading models. For example, a recent study by Ding et al. (2021) designed VIX-based trading strategies based on arbitrage theory. They find that VIX-based trading strategies can be used to exploit short-term return momentum and generate excess returns.[4] Specifically, most prior studies that reported positive profits calculate the TRs on the VIX to predict future price movements in the SPX market (e.g., Kozyra and Lento 2011; Zhu et al. 2019). The findings extend this stream of literature by revealing that positive profits are not consistently generated when simultaneously estimating TRs and trading in the VIX markets. Therefore, following the above literature, it appears that historical VIX data are more appropriately employed in a technical trading strategy in the SPX market than trading ETFs tied to the VIX.

The paper also extends the literature that explores the trading efficacy of the BBs. Lento et al. (2007) first explored the investment information content in BBs and found that the BBs were unable to generate positive profits consistently. However, they found that the BBs were consistently more profitable when calculated with a contrarian approach. More recently, Ni et al. (2020) explored the profitability of the BBs in the Taiwan stock market. They found that BBs are a profitable trading strategy, with additional profits being made when taking a contrarian approach only when share prices hit the upper BB. However, a study by Fang et al. (2017) suggested that the profitability of the BBs has decreased over time since their introduction in 1983. They conjectured that as trading rules become more popular, excess returns disappear, which is the case for the BBs after John Bollinger published his book in 2001.

The study reveals that the BBs continue to be among the most profitable trading rules commonly used by practitioners (see Table 2), even though their overall profitability may have declined over the past twenty years (Fang et al. 2017). Furthermore, the results reveal that during the market crash regime, it must be noted that at least one BB variant could generate statistically significant profits before transaction costs in the BTC, GLD, SPX, and VIX markets (see Appendix C). This provides novel insights into the BBs' ability to generate profits under more normal market conditions (e.g., Lento et al. 2007; Ni et al. 2020) and during market turbulence and uncertainty.

Regarding the three market regimes analyzed (i.e., normal market, market crash, and market recovery), it is also noted that the TRs were generally more profitable during

the market crash regime (see Appendix C). For example, statistically significant profits after transaction costs were generated by at least one variant of each trading rule on the SPX during the market crash. These findings suggest that the SPX market was not weak-form efficient during the market crash. Therefore, TRs could be employed as part of a portfolio management strategy during times of crisis. In addition, these findings stress the importance of trends and trading ranges that played a significant role during the "pandemic trading rounds". In other words, prices were driven by trends and momenta that were frequently interrupted by violent and swift adjustments of imbalances between supply and demand.

To further analyze the performance of the TRs during the market crash regime, the paths taken by a variant of each trading rule relative to the SPX (i.e., BHTS) are visualized. Figure 1 reveals that all four TR variants outperformed the BHTS. Specifically, the TRBO (50) and MACO (1,150) are shown to take very similar paths by generating positive profits but experiencing significant declines themselves. As a result, their positive profits are mainly driven by their ability to mitigate the extent of losses experienced by the SPX. The BB (20,1) and FR (2%) also performed similarly. However, their profits were associated with a much greater ability to avoid significant losses during the market crash regime. More precisely, the BB (20,1) and FR (2%) preserved wealth during the market crash regime. A large part of this wealth preservation is due to the BB (20,1) and FR (2%) avoiding some of the largest declines during the 16 March 2020 crash.

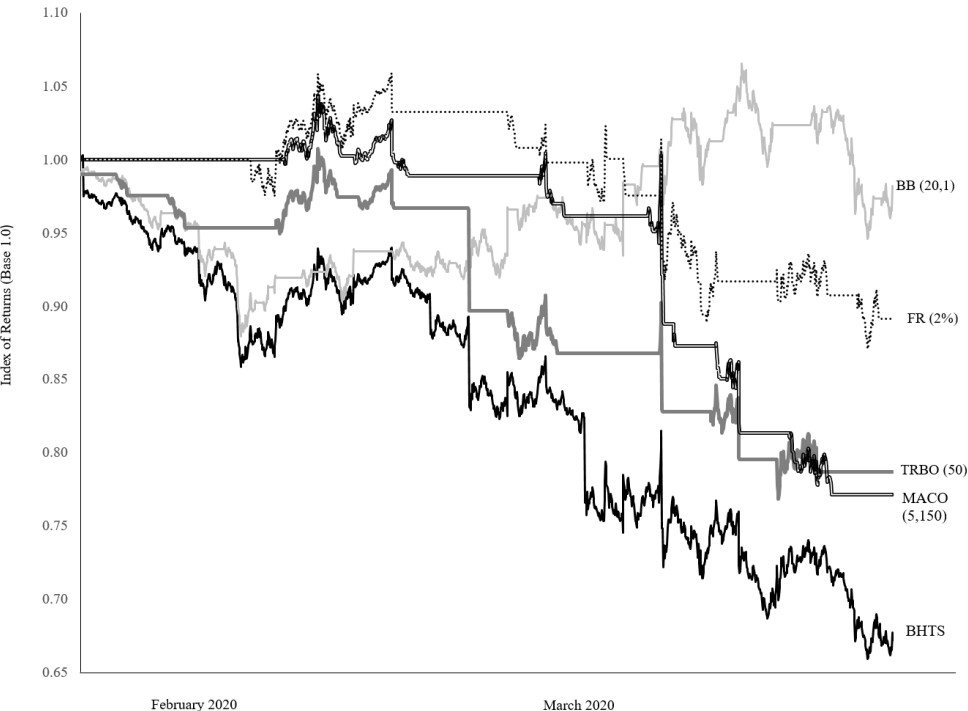

**Figure 1.** TTR Profitability relative to the BHTS on SPX during the market crash. Figure 1 presents the path taken by a variant of each trading rule relative to the SPX market (i.e., BHTS) during the market crash regime. The x-axis is the five-min interval data beginning 21 February 2020, and ending on 23 March 2020. The y-axis presents the index of returns with a base of 1.0 at the start of the market crash regime. Note that the start of the figure does not coincide with the start of the market crash regime, as several observations are required to calculate the trading rules.

Second, it is worthwhile to note that at least one MACO, FR, and TRBO variant all generated statistically significant profits after transaction costs on the GLD market during the market crash regime (see Appendix C). In addition, the BBs were able to generate statistically significant profits before transaction costs. These findings suggest that, although the GLD market may be efficient under normal market conditions, as suggested by

prior studies (e.g., Baur et al. 2020) and the sub-period results for the normal market regime (See Appendix C), excess returns may be available during periods of market distress.

These findings are important for portfolio managers as gold is often seen as a safe-haven investment based on its historical performance (e.g., Buccioli and Kokhol 2021; Areal et al. 2015). For example, during the aftermath of the 2008 global credit crisis, gold prices in USD more than tripled from August 2007 to August 2011. However, some have questioned gold's safe-haven properties during the COVID-19 pandemic. Recently, mainstream media has questioned whether gold has lost its luster as a safe-haven asset since it experienced a significant price decline of approximately 12% from 9 March 2020 to 18 March 2020, which was a lagged decline following the earlier and more pronounced drop in the SPX. Academic research by Akhtaruzzaman et al. (2021) corroborated these observations by suggesting that gold was a useful safe-haven asset during the early phase of the COVID-19 market meltdown (31 December 2019, to 16 March 2020) but lost its safe-haven status shortly after that (17 March 2020, to 24 April 2020). Regardless, Akhtaruzzaman et al. (2021) suggest that investors continued to invest in gold during the COVID-19 market meltdown.

These findings also suggest that TRs could be employed in the GLD market to devise active trading strategies based on TRs, as opposed to a more passive approach that utilizes the protective abilities of a gold hedge during a financial crisis. As discussed, TRs were especially useful in the GLD market during the market crash regime (see Appendix C), although not statistically significant across the entire period (see Table 2). To further analyze these results, the impact of the TRs in the GLD market is visualized. Specifically, Figure 2 presents the path of the SPX, GLD, and TRBO (200) in the GLD market across the entire sample period. The results reveal that the TRs successfully timed an appropriate exit from the GLD market before the significant drop during the market decline regime.

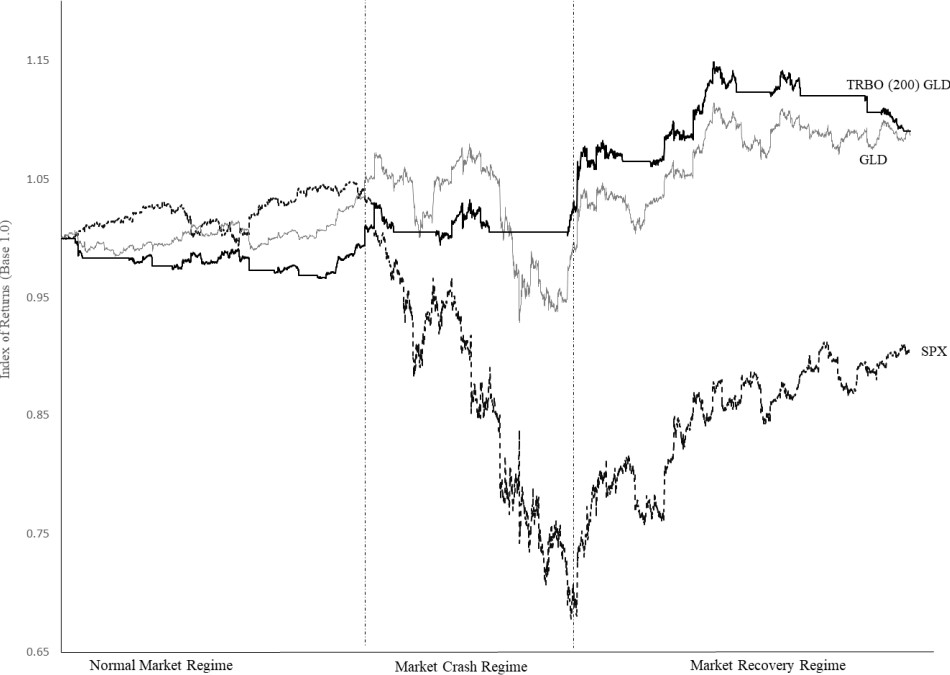

**Figure 2.** SPX, GLD, and TR pathways. Figure 2 presents the paths taken by the SPX, GLD, and the TRBO (200) trading rule (after taxes) in the GLD market across the entire period analyzed and three market regimes (i.e., Normal Market Regime, Market Crash Regime, and Market Recovery Regime). The x-axis presents the period of 6 January 2020 (i.e., the first day the TRBO (200) generated a trading signal) to 12 May 2020. The y-axis presents the index of returns with a base of 1.0 at the start of 6 January 2020.

The trading profits in the OIL market merit further discussion and analysis. Specifically, the TRs generated profits after transaction costs across nine of the twelve trading rule

variants on the full sample (see Table 2). The analysis of the results across the sub-periods supports the robustness of the findings across the entire sample as TRs were generally profitable across all three market regimes. Most interesting is that, although the TRs were profitable across the whole data span, their statistical significance was only evident during the market recovery regime (see Appendix C).

The individual five-minute interval returns across the entire sample is shown to further analyze these findings. Figure 3 reveals that the OIL market's vast majority of extreme volatility occurred during the SPX market's recovery period. As previously discussed, the OIL market experienced a major crash during the COVID-19 pandemic, partly driven by a price war between Saudi Arabia and Russia initiated on 8 March 2020, which eventually led oil futures to turn negative in April 2020. This price war occurred during the SPX market's recovery phase and can partially explain all of the volatility presented in Figure 3. During this period, the statistically significant excess returns resulted from the TRs being profitable during sharp downward movements in oil prices. Again, these findings suggest that the TRs may be useful during times of crisis as part of portfolio management strategies.

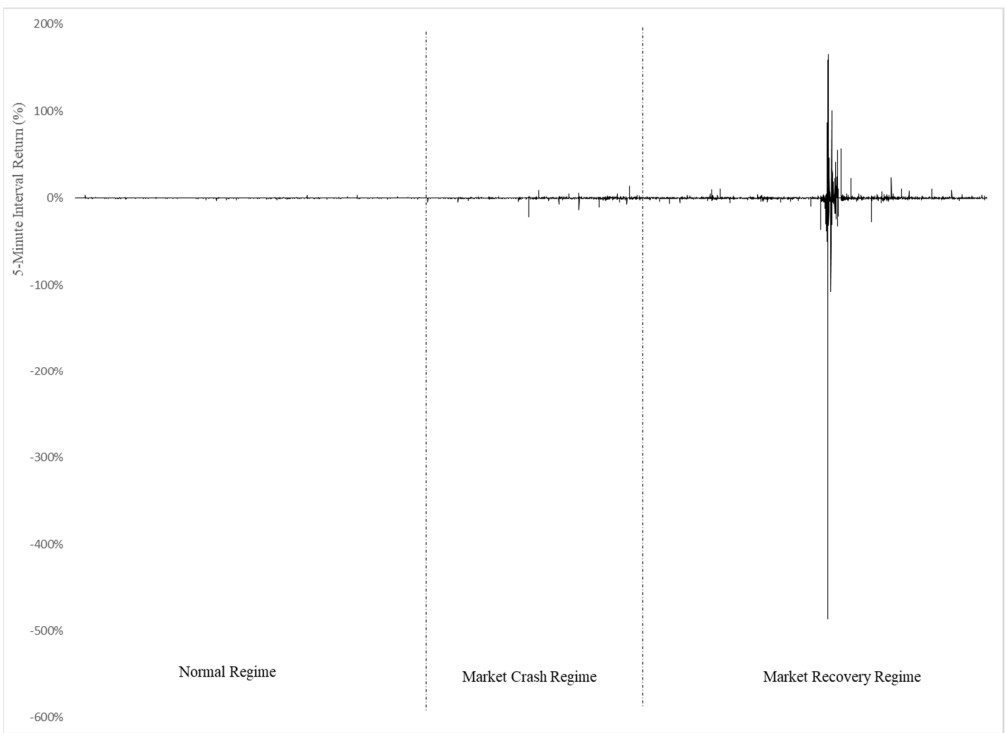

**Figure 3.** Individual 5-min interval returns in the OIL market. Figure 3 presents each five-minute interval return in the OIL market across the sample period and three market regimes (i.e., Normal Market Regime, Market Crash Regime, and Market Recovery Regime). The x-axis presents the period of 1 January 2020, to 12 May 2020. The y-axis presents the 5-min interval return percentage.

## 6. CSA Trading Strategy

A potential limitation of studies testing TRs on historical data sets is that a trader would not know which TR variant to rely upon, ex-ante. Furthermore, the study explores the profitability of the TRs across three market regimes. Again, a trader may not know, ex-ante, the beginning and end of the market crash and recovery regimes. Technical analysts could mitigate such limitations by aggregating the individual TR variants into a composite trading indicator employed equally across all market regimes.

A composite (or consensus) approach relies upon the concept of the "wisdom of crowds," which dates back to the early works of Francis Galton (1822–1911). The wisdom of crowds, in its simplest form, is a phenomenon whereby the average of a large number of independent estimates regarding a given question can be very close to the actual answer

(population mean), and the average is consistently more accurate than each estimate over time (Hertwig 2012; Surowiecki 2005). The wisdom of crowds concepts have been increasingly being incorporated into finance research (e.g., Chau et al. 2020; Chalmers et al. 2013; Ray 2006) and have already been employed with technical trading strategies (e.g., Lento and Gradojevic 2007; Wang et al. 2014; Ni et al. 2015).

The combined signal approach ("CSA") trading strategy is followed as put forward by Lento and Gradojevic (2007) by aggregating the TR variants into a composite signal and only going long when at least half or more of the 12 TR variants all agree on a buy or sell signal. The CSA trading strategy is explored when six (CSA (6/12)), seven (CSA (7/12)), eight (CSA (8/12)), nine (CSA (9/12)), ten (CSA (10/12)), and eleven (CSA (11/12)) trading signals agree. The analysis does not include anything below six as this would not suggest an average consensus on the trading signals. In addition, the approach does not test the CSA when all 12 TR variants agree as this would generate too few (if any) trading signals, especially considering the low number of trades generated by the FR variants in certain markets (see Appendix A).

Table 3 presents the results of the CSA trading strategy employed across the entire sample data with the twelve TR variants. The results reveal that the CSA (6/12), CSA (7/12), and CSA (8/12) are all able to generate profits before transaction costs across all five markets consistently. This suggests that the CSA trading strategy has some predictive ability above the naïve BHTS while reducing a trader's reliance upon any single TR variant across a given time. However, profits after transaction costs are observed in the BTC and WTI markets. The CSA trading approach was the least useful in the GLD and VIX markets, whereas forecasting ability was evident in the BTC, SPX, and WTI markets.

**Table 3.** CSA strategy profits across the entire sample period.

|  | BTC | GLD | SPX | VIX | WTI |
|---|---|---|---|---|---|
| CSA (6/12) | **0.22%**<br>**0.01%** | **0.05%**<br>−0.21% | **0.26%**<br>−0.05% | −0.41%<br>−0.94% | **2.96%**<br>**2.66%** |
| CSA (7/12) | **0.20%**<br>**0.05%** | **0.07%**<br>−0.19% | **0.23%**<br>−0.11% | **0.24%**<br>−0.20% | **0.69%**<br>**0.43%** |
| CSA (8/12) | **0.33%**<br>**0.09%** | **0.00%**<br>−0.29% | **0.15%**<br>−0.13% | **0.24%**<br>−0.42% | **1.34%**<br>**1.09%** |
| CSA (9/12) | **0.24%**<br>−0.07% | −0.04%<br>−0.33% | **0.23%**<br>**0.02%** | −0.87%<br>−1.54% | **1.10%**<br>**0.80%** |
| CSA (10/12) | −0.04%<br>−0.31% | −0.09%<br>−0.17% | **0.09%**<br>−0.04% | −0.80%<br>−0.95% | **0.62%**<br>**0.50%** |
| CSA (11/12) | −0.16%<br>−0.31% | −0.09%<br>−0.10% | **0.11%**<br>**0.05%** | −0.74%<br>−0.81% | **0.63%**<br>**0.60%** |

Notes: Table 3 presents the daily profits before and after transaction costs that are located in each cell above and below, respectively, across the entire sample period. Bold font represents trading profits whereby the trading rule returns exceeded the returns from the naïve buy-and-hold trading strategy (BHTS). The Combined Signal Approach (CSA (a/b)) involves going long when at least "a" or more of the "b" TR variants all agree on a buy or sell signal.

As an additional analysis, the CSA strategy is employed on each of the three market regimes (See Appendix D). The results reveal that the CSA strategy was very profitable during the market crash regime. These findings are consistent with the previous discussion of Figure 2, whereby the CSA beat the naïve buy-and-hold trading strategy during the market crash regime by preserving wealth by correctly timing market exits. Comparing the results across the market meltdown regime of the CSA trading strategy with the individual TRs reveals that the CSA is successful in reducing reliance upon any single trading rule and generated profits more consistently across the BTC, GOLD, OIL, and SPX markets.

The findings for BTC require more care and discussion. First, this work is related to that of Gerritsen et al. (2020) in that they tested trend-following indicators, while the current paper uses both trend and momentum TRs. Although the sample of Gerritsen et al. (2020) consisted of pre-COVID-19 daily data on BTC, their conclusions also indicated that the

trading range break-out rules were able to deliver superior performance in trending markets. In the same vein, Bouri et al. (2021) focused on pre-COVID-19 five-minute BTC data, but they did not employ TRs and used functional data analysis instead. Their results denied the weak form of the efficient market hypothesis, which is consistent with the main message from the current paper. Furthermore, when aided with sophisticated deep-learning methods, technical trading models that explore high-frequency predictability in the BTC market, such as those from Alonso-Monsalve et al. (2020), reinforced the evidence of departures from the efficient market hypothesis. However, Alonso-Monsalve et al. (2020) did not assess the economic value of their forecasts and their analysis covered the pre-COVID-19 time period (2018–2019). In all, the current paper complements the above literature and extends their samples to a more recent period that was characterized by abrupt regime shifts due to the emergence of the COVID-19 pandemic.

## 7. Conclusions

The main findings can be summarized as follows: (1) The financial crisis caused by the COVID-19 pandemic offers limited TR excess profits relative to the buy-and-hold strategy. (2) Trs that are statistically significantly profitable for almost all asset classes during the market crash (after accounting for transaction costs) are the Bollinger Bands and trading range break-out rules. (3) This suggests that market trend and momentum, impacted by the large imbalances between supply and demand, played a significant role in price formation and trader behavior in all asset classes. (4) Combined (or composite) CSA technical trading strategies can generate profitability improvements for all asset classes and are highly effective during the market crash regime.

Therefore, overall, the findings suggest that TRs may be useful for investors during market crashes and that many markets were weak-form inefficient during the COVID-19 market meltdown. These findings are consistent with the notion that, at such times, investors may become distressed by market trends and momentum and, consequently, shift from fundamental analysis to technical analysis. As a result, TRs and the resulting composite trading signals could play an essential role in a portfolio management strategy that seeks to preserve capital during times of distress. Moreover, the evidence of violations of the efficient market hypothesis could pose challenges to policymakers and regulators. In particular, if financial markets become inefficient in their weak form, then policy prescriptions to remedy the situation may not be clear and effective since the underlying causes for the inefficiency are unknown. However, it is important to note that because the Sharpe ratio does not account for all systematic risk, any findings of profitable TRs may pick up varying levels of systematic risk (and systematic risk premiums required by investors) and thus not indicate violations of the weak form of market efficiency.

The presented findings represent the first steps in understanding high-frequency, non-fundamental forces that were at play in the first months of the COVID-19 pandemic. To address the potential limitation of the use of a relatively small set of TRs, in the future, it would be helpful to expand the panel of technical indicators to those that, in addition to market trend and momentum, reflect the intraday market volume and aggregate sentiment of the investors (i.e., traders). Such measures may include, for instance, on-balance-volume, fear and greed, and other global indicators that could serve as proxies for the world risk appetite. In a recent paper, Makarov and Schoar (2020) demonstrated the importance of net order flows for price formation in cryptocurrency markets. Since order flows for assets effectively capture the information spreading process, they potentially contain both technical (behavioral) and fundamental sources of disruptions at the market microstructure level. Pending data availability at high frequencies, such a rich set of predictors might provide additional insights into profitable trading strategies during market distress. Another limitation of the current paper lies in its direct application of TRs. This approach could be extended to a full-fledged dynamic machine learning model that would rely on a large panel of TRs as its inputs (in the spirit of, e.g., Gradojevic et al. 2021). In addition, such complex models might be enriched by incorporating the skewness feature of

technical analysis as in Jin (2021). Finally, researchers could further extend the work of Bettman et al. (2009) to incorporate elements of both fundamental and technical analysis during the COVID-19 market meltdown to determine if there are any additional benefits for portfolio management.

**Author Contributions:** Conceptualization, C.L. and N.G.; methodology, C.L. and N.G.; software, C.L.; validation, C.L.; formal analysis, C.L.; investigation, C.L. and N.G.; resources, C.L. and N.G.; data curation, N.G.; writing—original draft preparation, C.L. and N.G.; writing—review and editing, C.L. and N.G.; visualization, C.L. and N.G.; supervision, C.L. and N.G.; project administration, C.L. and N.G. All authors have read and agreed to the published version of the manuscript.

**Funding:** This research received no external funding.

**Institutional Review Board Statement:** Not applicable.

**Informed Consent Statement:** Not applicable.

**Data Availability Statement:** Not applicable.

**Conflicts of Interest:** The authors declare no conflict of interest.

## Appendix A. Number of Trade Signal Generated by TR

**Table A1.** Number of Trading Signals Generated by TR.

| | BTC | GLD | SPX | VIX | WTI |
|---|---|---|---|---|---|
| MACO (1,50) | 475 (146/109/212) | 458 (159/91/207) | 456 (164/114/175) | 559 (218/133/204) | 427 (156/93/169) |
| MACO (1,200) | 133 (44/25/43) | 165 (32/34/99) | 221 (97/56/66) | 261 (128/53/63) | 174 (62/43/57) |
| MACO (5,150) | 99 (22/24/45) | 108 (30/32/45) | 119 (47/26/44) | 135 (65/33/33) | 90 (36/21/30) |
| BB (20,2) | 585 (189/134/251) | 585 (185/133/264) | 579 (201/142/231) | 574 (219/132/219) | 525 (193/121/204) |
| BB (20,1) | 766 (269/173/305) | 765 (242/185/333) | 703 (254/182/261) | 736 (288/177/267) | 697 (262/169/254) |
| BB (30,2) | 477 (158/115/189) | 478 (156/108/211) | 422 (157/110/150) | 474 (175/122/172) | 428 (167/94/159) |
| FR (1%) | 278 (60/123/82) | 50 (2/37/10) | 223 (3/164/55) | 1,447 (408/574/464) | 765 (36/201/527) |
| FR (2%) | 112 (19/57/28) | 9 (0/7/1) | 21 (0/49/16) | 729 (169/389/170) | 330 (8/85/236) |
| FR (5%) | 21 (2/11/6) | 0 (0/0/0) | 9 (0/8/0) | 170 (26/117/27) | 94 (0/17/77) |
| TRBO (50) | 447 (174/106/107) | 501 (164/58/127) | 557 (208/38/55) | 501 (165/65/69) | 468 (149/39/527) |
| TRBO (150) | 254 (100/52/70) | 306 (104/34/75) | 323 (124/6/140) | 271 (80/45/22) | 283 (89/8/91) |
| TRBO (200) | 226 (84/46/84) | 272 (90/28/106) | 287 (111/2/93) | 229 (65/40/107) | 265 (81/5/52) |

Note: Table A1 presents the number of trading signals generated by each trading rule variant across each of the five markets (i.e., BTC, GLD, SPX, VIX, and WTI). The total number of signals generated across the entire sample is presented above, while the number of signals generated across each market regime is presented below (i.e., normal market regime, market crash regime, and market recovery regime).

## Appendix B. TTR Sharpe Ratios across the Full Sample

**Table A2.** TTR Sharpe Ratios across the full sample.

| | SRs for the MACO TTRs | | | | | |
|---|---|---|---|---|---|---|
| | MACO (1,50) | | MACO (1,200) | | MACO (5,150) | |
| | Before Tx Costs | After Tx Costs | Before Tx Costs | After Tx Costs | Before Tx Costs | After Tx Costs |
| BTC | −1.719 | −4.097 | −1.248 | −2.111 | −2.125 | −2.575 |
| GLD | −4.110 | −5.785 | 0.698 | −8.889 | 0.640 | −6.270 |
| OIL | 3.151 | 4.027 | 9.112 | 6.560 | 2.328 | 2.527 |
| SPX | −1.632 | −4.458 | −0.305 | −3.061 | 0.517 | −1.635 |
| VIX | −11.494 | −14.189 | −2.533 | −4.653 | −1.801 | −2.863 |
| | SRs for the BB TTRs | | | | | |
| | BB (20,2) | | BB (20,1) | | BB (30,2) | |
| | Before Tx Costs | After Tx Costs | Before Tx Costs | After Tx Costs | Before Tx Costs | After Tx Costs |
| BTC | 1.367 | −3.916 | 1.657 | −5.323 | 0.995 | −3.868 |
| GLD | 0.245 | −6.345 | 0.149 | −6.095 | −3.115 | −6.461 |
| OIL | −3.334 | −4.633 | −0.826 | −2.978 | −2.946 | −3.928 |
| SPX | 4.890 | −2.258 | 4.058 | −3.226 | 4.820 | −2.005 |
| VIX | 0.394 | 0.383 | 0.398 | 0.326 | 0.412 | 0.305 |
| | SRs for the FR TTRs | | | | | |
| | FR (1%) | | FR (2%) | | FR (5%) | |
| | Before Tx Costs | After Tx Costs | Before Tx Costs | After Tx Costs | Before Tx Costs | After Tx Costs |
| BTC | −1.598 | −2.387 | −1.083 | −1.471 | 0.226 | 0.081 |
| GLD | −1.729 | −3.238 | −0.274 | −0.382 | 0.000 | 0.000 |
| OIL | 2.322 | 3.120 | 1.192 | 1.419 | 1.260 | 1.333 |
| SPX | 1.308 | −0.719 | 4.608 | 3.829 | 4.514 | 4.550 |
| VIX | −1.391 | −12.266 | −6.640 | −10.814 | −2.388 | −2.820 |
| | SRs for the TRBO TTRs | | | | | |
| | TRBO (50) | | TRBO (150) | | TRBO (200) | |
| | Before Tx Costs | After Tx Costs | Before Tx Costs | After Tx Costs | Before Tx Costs | After Tx Costs |
| BTC | −1.009 | −1.488 | 1.688 | 1.544 | 1.107 | 0.862 |
| GLD | 0.863 | −1.965 | −0.067 | −1.272 | 0.576 | 0.044 |
| OIL | 3.310 | 3.507 | 11.604 | 11.083 | 7.292 | 6.757 |
| SPX | 1.563 | −0.779 | 1.688 | 0.866 | 5.103 | 4.415 |
| VIX | −4.850 | −6.415 | −0.618 | −0.721 | 0.264 | 0.237 |

Note: Table A2 presents the SRs before and after transaction costs for each TTR variant. The SRs are calculated in accordance with Equation (5). Following Zhu et al. (2015), we present the SR after being multiplied by a factor or $10^2$. Note that higher SRs indicate that a TTR generated higher mean returns and/or less volatility relative to TTRs with lower SRs.

## Appendix C. TTR Profitability across COVID-19 Market Meltdown Market Regimes

**Table A3.** TTR Profitability across COVID-19 Market Meltdown Market Regimes.

**TTR Profitability across the Normal Market Regime**

| | MACO (1,50) | MACO (1,200) | MACO (5,150) | BB (20,2) | BB (20,1) | BB (30,2) | FR (1%) | FR (2%) | FR (5%) | TRBO (50) | TRBO (150) | TRBO (200) |
|---|---|---|---|---|---|---|---|---|---|---|---|---|
| BTC | −0.73%<br>−1.36% | −0.44%<br>−0.64% | −0.45%<br>−0.55% | −0.49%<br>−0.81% | −0.34%<br>−0.91% | −0.61%<br>−0.87% | −0.29%<br>−0.42% | −0.44%<br>−0.48% | −0.74%<br>−0.75% | −0.54%<br>−0.67% | −0.36%<br>−0.40% | −0.61%<br>−0.64% |
| GLD | −0.06%<br>−0.75% | −0.01%<br>−0.16% | **0.02%**<br>−0.12% | −0.08%<br>−0.47% | −0.14%<br>−0.69% | −0.13%<br>−0.43% | −0.03%<br>−0.04% | −0.16%<br>−0.16% | −0.16%<br>−0.16% | −0.07%<br>−0.19% | −0.08%<br>−0.13% | −0.09%<br>−0.13% |
| OIL | −0.01%<br>−0.69% | **0.33%**<br>**0.04%** | **0.36%**<br>**0.20%** | **0.39%**<br>−0.01% | **0.52%**<br>−0.09% | **0.53%**<br>**0.21%** | **0.08%**<br>**0.01%** | **0.36%**<br>**0.35%** | **0.42%**<br>**0.42%** | **0.06%**<br>−0.08% | **0.48%**<br>**0.45%** | **0.47%**<br>**0.45%** |
| SPX | −0.10%<br>−0.82% | −0.08%<br>−0.53% | −0.04%<br>−0.26% | **0.04%**<br>−0.40% | **0.06%**<br>−0.63% | −0.01%<br>−0.36% | −0.01%<br>−0.01% | −0.13%<br>−0.13% | −0.13%<br>−0.13% | −0.12%<br>−0.29% | −0.08%<br>−0.13% | −0.07%<br>−0.10% |
| VIX | −1.89%<br>−2.82% | −1.30%<br>−1.88% | −0.82%<br>−1.11% | **1.81% ***<br>**1.30% *** | **1.95% ***<br>**1.16% *** | **1.22%**<br>**0.82%** | −1.26%<br>−2.37% | −0.84%<br>−1.23% | −0.70%<br>−0.75% | −1.62%<br>−1.82% | −0.46%<br>−0.51% | −0.34%<br>−0.37% |

**TTR Profitability across the Market Crash Regime**

| | MACO (1,50) | MACO (1,200) | MACO (5,150) | BB (20,2) | BB (20,1) | BB (30,2) | FR (1%) | FR (2%) | FR (5%) | TRBO (50) | TRBO (150) | TRBO (200) |
|---|---|---|---|---|---|---|---|---|---|---|---|---|
| BTC | −0.17%<br>−0.85% | **0.38%**<br>**0.19%** | **0.38%**<br>**0.12%** | **2.17% *** <br>**1.78% *** | **2.12% *** <br>**1.60% *** | **2.18% *** ***<br>**1.84% *** *** | −0.44%<br>−0.95% | **0.00%**<br>−0.23% | **2.10% *** ***<br>**2.08% *** | −0.57%<br>−0.72% | **1.85% *** ***<br>**1.82% *** *** | **1.68% *** <br>**1.66% *** |
| GLD | **0.17%**<br>−0.39% | **0.44% *** <br>**0.21% *** | **0.44% *** <br>**0.18% *** | **0.24% ***<br>−0.13% | **0.27% ***<br>−0.33% | **0.13%**<br>−0.17% | **0.00%**<br>−0.11% | **0.19%**<br>**0.18%*** | **0.15%**<br>**0.15%** | **0.28% *** <br>**0.15% *** | **0.10%**<br>**0.04%** | **0.50% *** <br>**0.47% *** |
| OIL | 2.67%<br>2.12% | 2.34%<br>2.06% | 2.34%<br>2.62% | 1.42%<br>1.10% | 3.09%<br>2.57% | 1.69%<br>1.45% | 1.94%<br>1.23% | 3.71%<br>3.47% | 3.19%<br>3.15% | 3.14%<br>3.05% | 3.14%<br>3.11% | 2.92%<br>2.89% |
| SPX | −0.02%<br>−0.69% | **0.47%**<br>**0.09%** | **0.47%**<br>**0.57% *** | **1.96% *** ***<br>**1.59% *** *** | **2.00% *** ***<br>**1.45% *** *** | **1.55% *** <br>**1.29% *** *** | **0.37%**<br>−0.19% | **1.36% *** <br>**1.21% *** | **1.46% *** <br>**1.45% *** | **0.79% *** <br>**0.66% *** | **0.89% ***<br>**0.85% *** | **1.37% *** ***<br>**1.36% *** *** |
| VIX | −8.45%<br>−9.23% | −3.80%<br>−4.17% | −3.80%<br>−4.10% | **1.61% ***<br>**1.18%** | **1.22%**<br>**0.62%** | −0.92%<br>−1.27% | −2.58%<br>−5.39% | −4.83%<br>−6.36% | −5.72%<br>−6.13% | −3.67%<br>−3.80% | −4.77%<br>−4.83% | −1.40%<br>−1.43% |

**TTR Profitability across the Market Recovery Regime**

| | MACO (1,50) | MACO (1,200) | MACO (5,150) | BB (20,2) | BB (20,1) | BB (30,2) | FR (1%) | FR (2%) | FR (5%) | TRBO (50) | TRBO (150) | TRBO (200) |
|---|---|---|---|---|---|---|---|---|---|---|---|---|
| BTC | −0.58%<br>−1.46% | −0.30%<br>−0.48% | −0.79%<br>−0.98% | −0.59%<br>−1.07% | −0.46%<br>−1.17% | −0.78%<br>−1.16% | −0.20%<br>−0.32% | −0.25%<br>−0.28% | −0.61%<br>−0.62% | −0.25%<br>−0.35% | −0.37%<br>−0.40% | −0.22%<br>−0.24% |
| GLD | −0.24%<br>−1.07% | −0.14%<br>−0.53% | −0.16%<br>−0.35% | **0.06%**<br>−0.42% | **0.10%**<br>−0.59% | **0.00%**<br>−0.33% | −0.13%<br>−0.15% | −0.12%<br>−0.13% | −0.17%<br>−0.17% | −0.01%<br>−0.14% | −0.09%<br>−0.14% | −0.06%<br>−0.10% |
| OIL | **4.19% *** <br>**3.47% *** | **1.54%**<br>**1.32%** | **6.05% *** <br>**5.92% *** | −3.89%<br>−4.25% | −3.29%<br>−3.81% | −3.84%<br>−4.08% | **5.22% ***<br>**3.72% *** | **6.81% *** <br>**6.16% *** | **7.10% *** <br>**6.49% *** | **3.61% ***<br>**3.49% *** | **1.20%**<br>**1.17%** | **0.17%**<br>**0.14%** |
| SPX | −0.27%<br>−0.98% | −0.30%<br>−0.59% | −0.28%<br>−0.46% | −0.49%<br>−0.86% | −0.37%<br>−0.90% | −0.53%<br>−0.78% | **0.09%**<br>−0.01% | −0.28%<br>−0.31% | −0.62%<br>−0.62% | −0.08%<br>−0.20% | −0.13%<br>−0.17% | −0.29%<br>−0.32% |
| VIX | **0.39%**<br>−0.38% | **1.32%**<br>**1.05%** | **1.69%**<br>**1.55%** | **1.09%**<br>**0.72%** | **0.91%**<br>**0.36%** | **1.05%**<br>**0.78%** | **1.05%**<br>−0.02% | **0.65%**<br>**0.27%** | **1.53%***<br>**1.49%** | **1.18%**<br>**1.03%** | **1.63%**<br>**1.58%** | **1.46%**<br>**1.44%** |

Note: Table A3 presents the daily profits before and after transaction costs are above and below, respectively. Bold font represents trading profits whereby the trading rule returns exceeded the returns from the naïve BHTS. ***, **, * represent statistically significant trading profits based on the Levich and Thomas (1993) bootstrapping technique at the 1%, 5%, and 10% levels, respectively.

## Appendix D. CSA Strategy Profits across the COVID-19 Market Meltdown Market Regimes

**Table A4.** CSA Strategy Profits across the COVID-19 Market Meltdown Market Regimes.

| | BTC | GLD | SPX | VIX | WTI |
|---|---|---|---|---|---|
| **TTR Profitability across the normal market regime** | | | | | |
| CSA (6/12) | −0.37% | −0.01% | 0.00% | −0.51% | **0.48%** |
| | −0.58% | −0.22% | −0.41% | −1.01% | **0.29%** |
| CSA (7/12) | −0.24% | −0.05% | −0.07% | −0.19% | **0.53%** |
| | −0.38% | −0.27% | −0.61% | −0.56% | **0.40%** |
| CSA (8/12) | −0.24% | −0.07% | −0.09% | **0.21%** | **0.46%** |
| | −0.36% | −0.39% | −0.49% | −0.41% | **0.34%** |
| CSA (9/12) | −0.86% | −0.04% | −0.15% | −0.58% | **0.59%** |
| | −1.04% | −0.34% | −0.28% | −1.11% | **0.45%** |
| CSA (10/12) | −0.91% | −0.10% | −0.15% | −0.10% | **0.58%** |
| | −1.14% | −0.14% | −0.16% | −0.18% | **0.52%** |
| CSA (11/12) | −0.92% | −0.10% | −0.15% | −0.11% | **0.54%** |
| | −1.16% | −0.10% | −0.15% | −0.12% | **0.51%** |
| **TTR Profitability across the market crash regime** | | | | | |
| CSA (6/12) | **1.73%** | **0.35%** | **1.25%** | −3.43% | **3.27%** |
| | **1.44%** | **0.01%** | **0.92%** | −4.18% | **2.93%** |
| CSA (7/12) | **1.68%** | **0.50%** | **1.17%** | −1.83% | **2.66%** |
| | **1.53%** | **0.28%** | **0.97%** | −2.64% | **2.38%** |
| CSA (8/12) | **1.51%** | **0.45%** | **1.17%** | −3.18% | **3.82%** |
| | **1.31%** | **0.27%** | **1.05%** | −4.53% | **3.68%** |
| CSA (9/12) | **2.14%** | **0.28%** | **1.64%** | −4.01% | **3.74%** |
| | **1.89%** | **0.14%** | **1.60%** | −5.36% | **3.64%** |
| CSA (10/12) | **1.86%** | **0.33%** | **1.66%** | −4.38% | **3.88%** |
| | **1.79%** | **0.32%** | **1.61%** | −4.75% | **3.85%** |
| CSA (11/12) | **1.97%** | **0.35%** | **1.68%** | −4.22% | **3.89%** |
| | **1.97%** | **0.35%** | **1.68%** | −4.44% | **3.89%** |
| **TTR Profitability across the market recovery regime** | | | | | |
| CSA (6/12) | −0.28% | −0.11% | −0.10% | **1.42%** | **5.70%** |
| | −0.40% | −0.42% | −0.39% | **1.01%** | **5.33%** |
| CSA (7/12) | −0.19% | −0.11% | −0.27% | **1.92%** | −0.17% |
| | −0.24% | −0.39% | −0.54% | **1.64%** | −0.47% |
| CSA (8/12) | −0.21% | −0.08% | −0.30% | **2.43%** | 0.84% |
| | −0.45% | −0.38% | −0.62% | **2.19%** | 0.45% |
| CSA (9/12) | −0.31% | −0.11% | −0.53% | **1.75%** | 0.22% |
| | −0.78% | −0.35% | −0.79% | **1.42%** | −0.29% |
| CSA (10/12) | −0.86% | −0.14% | −0.39% | **2.12%** | −1.28% |
| | −1.29% | −0.24% | −0.52% | **2.06%** | −1.49% |
| CSA (11/12) | −1.24% | −0.14% | −0.40% | **2.24%** | −1.24% |
| | −1.43% | −0.15% | −0.40% | **2.22%** | −1.29% |

Note: Table A4 presents whether the daily profits before and after transaction costs are above and below, respectively, across the three market meltdown regimes. Bold font represents trading profits whereby the trading rule returns exceeded the returns from the naïve BHTS.

## Notes

[1]   See and Nazário et al. (2017) and Neely and Weller (2012) for extensive surveys on the application of technical analysis in financial markets.

[2]   The returns calculated in this study are based on spot indices and therefore may not reflect a true return that would include components, such as a dividend yield (e.g., the S&P 500), convenience yield (e.g., gold), and holding cost (e.g., gold and oil commodities). Investors employing a trading strategy using actual futures contracts (or ETFs) would incorporate such components into their return measures.

[3]   We also calculated the percentage of the individual technical indicators that are significantly profitable (at both the 1-day and 10-day lags) to the percentage that would exist by chance assuming a random walk with a drift. The un-tabulated results reveal that the sell signals were more profitable than the buy signals for the VIX and OIL markets driven by the MACO, BB, and TRBO rules. However, buy signals were more profitable for the BTC and GLD mainly driven by the MACO trading rules. These results are not reported for brevity, but they can be available upon request from the authors. We thank the three anonymous referees and the Editor for this and other useful suggestions.

[4]   In a related paper, Xu et al. (2020) found predictability in the volatility indices of commodity exchange-traded funds, especially on days with higher volatility and larger jumps. It is important to note that volatility indices cannot be traded directly, but by constructing a portfolio of options that replicates the volatility index. Moreover, recently, Wang et al. (2022) showed that multiscale trading strategies based on the VIX may be possible.

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
