# Peer review of "The Profitability of Technical Analysis during the COVID-19 Market Meltdown"

_jrfm, doi:10.3390/jrfm15050192_

Round 1

Reviewer 1 Report

This article provides an interesting analysis of the effectiveness of technical analysis with high frequency data on a few indices in early 2020 when the pandemic began. The results indicate that the individual technical trading rules don't provide much opportunity for significant profits after transaction costs. However, the authors find better predictive performance when half or more of the indicators give the same signal.

There are several areas of improvement that could be carried out rather easily.

First of all, returns are measured largely on spot indices and therefore don't reflect a true return that would include the spot yield (like on the S&P 500) or the convenience yield and holding cost (for the commodities). A buy-and-hold strategy using actual futures contracts (or ETFs) would incorporate such (as well as market information in the basis that may be anything that the group of technical indicators may be picking up). This needs to be explained in the article.

Secondly, it would be useful to compare the percentage of the individual technical indicators that are significantly profitable to the percentage that would exist by chance at the various significance levels.

Thirdly, it should be explicitly stated in the paper that, because there is no adjustment for systematic risk, any finding of a profitable set of trading rules may be merely picking up varying levels of systematic risk (and systematic risk premiums required by investors) and thus not indicate anything about weak form market efficiency.

Reviewer 2 Report

JRFM: The Profitability of Technical Analysis during the COVID-19 Market Meltdown

The authors of this paper examine the ability of technical analysis to generate abnormal returns in various assets covering the S&P 500 index, Bitcoin, Comex gold spot, crude oil WTI, and the VIX, during the pandemic. To this end, they apply the Combined Signal Approach (CSA) and report some interesting results.

I like the idea of the paper given that predicting asset prices remain a relevant and appealing research questions and that implications for many economic actors, especially if the applied methods consider technical analysis and can be used to generate abnormal profits.  My minor comments are as follows. Once addressed carefully, I can reconsider my decision:

1- In the introduction section, the authors should spend more ink to better motivate the study and explain their added value compared to the existing literature. Maybe you should a small paragraph describing the contribution of the paper as compared to related studies.

2-         I would have liked to see how the empirical analysis and the key empirical findings presented in this paper are related to findings from previous papers. This would add more value to the analyses and make it more susceptible to citations from the rising literature on the predictability of asset prices. In this regard, you should consider some recent evidence on the ability of technical trading rules to generate abnormal profits in the Bitcoin market (see, “Convolution on neural networks for high-frequency trend prediction of cryptocurrency exchange rates using technical indicators. Expert Systems with Applications, 149, 113250”; “The profitability of technical trading rules in the Bitcoin market. Finance Research Letters, Vol. 34, 101263”; “On the intraday return curves of bitcoin: predictability and trading opportunities. International Review of Financial Analysis, Vol. 76, 101784”;

3- Try to add more economic intuition of the results, maybe regarding the efficiency/inefficiency of the financial markets. Furthermore, consider the discussion of results in light of the existing literature.

4- More explanations and justifications are needed regarding the following: “These findings also suggest that TRs could be employed in the GLD market to further enhance the safe-haven properties of gold during financial market duress ”.  How can you do this?

5- Regarding the following: “They find that VIX-based trading strategies can be used to exploit short-term return momentum and generate excess returns”, it would be nice to consider some recent evidence on the VIX and its scale-based dependence with other VIXs (see “Intraday return predictability: Evidence from commodity ETFs and related volatility indices. Resources Policy, Vol. 69, 101830”; “A grey-based correlation with multi-scale analysis: S&P 500 VIX and individual VIXs of large US company stocks. Finance Research Letters, https://doi.org/10.1016/j.frl.2022.102872”)

6-         In the conclusion section, a more detailed discussion of the policy implications would make the paper richer and more informative for investors, and traders. Also, consider describing the study limitations and scope for future research. In this regard, it would be interesting to point to the possibility of applying the methods in the universe of fiat currencies as in: “Jin, X. (2021). What do we know about the popularity of technical analysis in foreign exchange markets? A skewness preference perspective. Journal of International Financial Markets, Institutions and Money, 71, 101281”.

7-Make tables more self-explanatory.

Reviewer 3 Report

This is a very interesting paper with an important contribution to relevant research. However, some issues need to be addressed for its publication in Journal of Risk and Financial Management.

Main comments and Suggestions for Authors:

Research should be written impersonally. Therefore, I recommend that you remove the "we" or "our" from the entire article. For example (L.9, L.14, L.90, L.91, L.125, L.134....)

It would be ideal if the superscript 1 was on the page where it is mentioned, that is, on the first page. It may be more difficult to find it if it is included at the end of the article.

Check the references and their style are according to the journal requirements. Please add the DOI of each investigation considered.

The results and contributions obtained should not be put in the introduction. This information should be addressed at the end of the article when the main findings are presented.

Author Response

This manuscript is a resubmission of an earlier submission. The following is a list of the peer review reports and author responses from that submission.

Round 1

Reviewer 1 Report

The paper examines the performance of several simple trading signals in several assets. Overall, I understand the idea of the paper, but the contribution of the paper is minimal. I can hardly see how it may attract citations in the future. It looks more like an applied research paper. Furthermore, the quality of empirical analysis is generally low. The data sample is small and has substantial deficiencies. The examinations lack a proper statistical toolset, accounting for risk, inferences, etc. In consequence, I recommend the rejection of the paper.

Further comments:
1. How do the authors precisely account for the transaction costs?
2. Are all of the securities used in the study tradeable? Do they represent actual attainable returns? For example, the paper does not specify whether the S&P index is price-based or return-based. A price-based index cannot be traded; it is just an indicator.
3. The evaluation of the strategies is based purely on returns. It would be beneficial to adjust it for risk, compare it with some benchmarks, etc.
4. The presentation of the results lack clarity.

Reviewer 2 Report

The article covers and interesting topic of trading strategies based on technical analysis performance during Covid-19 pandemic. It is quite well written and the conclusions are supported by the provided results.
I found the following imperfections:

1. To be precise, not the WTI oil prices become negative, but the WTI futures on the next expiry date.

2. The big tables 2,3,4 should be moved to appendix or compressed into one with the most significant results.

3. Figs 1,2 - no y-axis description.

4. In addition to the discussion presented in lines 314-326 regarding the gold safe haven role, it should be mentioned that the Covid-19 pandemic period was quite special - almost all assets expressed in USD, including gold were strongly correlated.  - see recent paper: "Cryptocurrency Market Consolidation in 2020–2021", Entropy 2021, 23(12), 1674; https://doi.org/10.3390/e23121674

Reviewer 3 Report

In this paper, the authors propose to analyse the profitability of technical trading rules in several assets, during the COVID-19, with a sample from January to May 2020, considering high-frequent data. I believe that the paper has several problems:

  1. Considering the theme under analysis, it needs clearly a literature review, which does not exist. Moreover, statements like the one in lines 23/24 are not consistent, once it is possible to find many and many works.
  2. Another concern is the fact that the paper has a dataset totally out of date and with a small time sample. I understand that the use of high-frequent data makes the statistical analysis possible, due to the existence of high number of observations, although the sample does not allow to retrieve conclusions. In particular, this is the main issue that I find in the paper, implying that I do not consider the paper proper to be considered for publication,